# Methods to Identify Cognitive Alterations from Animals to Humans: A Translational Approach

**DOI:** 10.3390/ijms24087653

**Published:** 2023-04-21

**Authors:** Daniela Navarro, Ani Gasparyan, Silvia Martí Martínez, Carmen Díaz Marín, Francisco Navarrete, María Salud García Gutiérrez, Jorge Manzanares

**Affiliations:** 1Instituto de Neurociencias, Universidad Miguel Hernández-CSIC, Avda de Ramón y Cajal s/n, San Juan de Alicante, 03550 Alicante, Spain; 2Redes de Investigación Cooperativa Orientada a Resultados en Salud (RICORS), Red de Investigación en Atención Primaria de Adicciones (RIAPAd), Instituto de Salud Carlos III, MICINN and FEDER, 28029 Madrid, Spain; 3Instituto de Investigación Sanitaria y Biomédica de Alicante (ISABIAL), 03010 Alicante, Spain; 4Servicio de Neurología, Hospital General Dr. Balmis, 03010 Alicante, Spain

**Keywords:** human cognitive trials, dementia, cognitive impairment, memory, animal models

## Abstract

The increasing prevalence of cognitive dysfunction and dementia in developed countries, associated with population aging, has generated great interest in characterizing and quantifying cognitive deficits in these patients. An essential tool for accurate diagnosis is cognitive assessment, a lengthy process that depends on the cognitive domains analyzed. Cognitive tests, functional capacity scales, and advanced neuroimaging studies explore the different mental functions in clinical practice. On the other hand, animal models of human diseases with cognitive impairment are essential for understanding disease pathophysiology. The study of cognitive function using animal models encompasses multiple dimensions, and deciding which ones to investigate is necessary to select the most appropriate and specific tests. Therefore, this review studies the main cognitive tests for assessing cognitive deficits in patients with neurodegenerative diseases. Cognitive tests, the most commonly used functional capacity scales, and those resulting from previous evidence are considered. In addition, the leading behavioral tests that assess cognitive functions in animal models of disorders with cognitive impairment are highlighted.

## 1. Introduction

In medicine, cognitive function disorders are generally associated with “primary degenerative dementias,” particularly important because of their high frequency and prevalence in the ageing populations typical of developed countries. This heterogeneous group of diseases is characterized by progressive mental deterioration, including memory loss and/or impaired judgment, calculation, language, orientation, skills and behavior, and psychiatric disorders. Anatomopathological findings show neuronal loss, especially in the cerebral cortex, along with degenerative changes in the surviving neurons, protein inclusions, and vascular and glial alterations. The location of neuronal loss and the nature of the degenerative changes and inclusions are the factors that differentiate the various conditions [1]. Examples of this group of degenerative diseases include frontotemporal dementia, dementia with diffuse Lewy bodies, and Alzheimer’s disease [2]. In addition, there are cognitive alterations in other degenerative neurological diseases whose cardinal manifestations consist of movement disorders, such as cerebellar ataxias, motor neuron diseases, muscular dystrophies, Parkinson’s disease [3,4], essential tremor [5], Huntington’s disease [6,7], progressive supranuclear palsy [8], hereditary ataxias [9], amyotrophic lateral sclerosis [10], and myopathies [11].

Notably, virtually all acquired or secondary neurological diseases of the nervous system, including traumatic, vascular, infectious, inflammatory, demyelinating, tumor, iatrogenic, toxic, metabolic, and psychiatric disorders have been associated with intellectual impairment or dementia [12]. Likewise, some systemic diseases, such as chronic pain [13] or rheumatic [14] and digestive [15] conditions, have been associated with cognitive dysfunction. Therefore, the ubiquity of intellectual dysfunction in human pathology and the fact that cognitive impairment is always a major cause of disability for the patient highlights the importance of the problem and the need to generate knowledge on the causes, pathogenetic mechanisms, and treatments of these disorders. Several tests are used to evaluate cognitive impairment in these patients.

Despite the remarkable progress on techniques and knowledge in neuroimaging, neurochemistry, neurogenetics, neuropsychology, neurophysiology, and neuropharmacology, research in humans presents significant limitations that make experimental studies with animal models of the different types of cognitive dysfunction essential. There is a wide variety of animal models of cognitive impairment, classified according to the mechanism of cognitive impairment, genetic selection or manipulation, environmental agents during development or adulthood, or pharmacological manipulation of the animal’s cognitive response. Although in vitro studies of brain tissue are of interest, in the field of cognitive pharmacology, it is necessary to use one or more animal models to determine the usefulness of a new drug and its dosage due to the complexity of cognitive function, in which multiple interconnected brain structures are involved.

For all the above reasons, this review describes the main cognitive tests used in humans to identify cognitive alterations and behavioral tests in experimental animals to assess a range of cognitive disorders, examining their rationale, advantages, and disadvantages (Table 1).

## 2. Cognitive Function: Attention, Learning, and Memory

The cognition system comprises functional, learning, and memory processes. Functional processes include “pre-attention” processes, in which the filtering of sensorimotor information is performed to focus attention on the most relevant elements of the environment [16] and whose alteration generates sensory overload and cognitive fragmentation that could contribute to the alterations that occur in different psychotic disorders [17]. On the other hand, “attention” processes are thoughtful attention, visual orientation, learned orientation, vigilance, habituation, and selective, sustained, or divided attention.

Learning and memory processes include associative, spatial, or non-spatial learning and short- and long-term memory. The concept of memory can be defined as “the long-term retention of experience-dependent internal representations” [18]. Memory can be divided into two major blocks: implicit (non-declarative) and explicit (declarative) memory [19,20]. Implicit memory is built from non-conscious learning processes through habits and skills, prior stimulation, or sensitization. It involves skeletal musculature or emotional responses and non-associative learning. Explicit or declarative memory involves retaining knowledge of certain events, places, or facts and implicates the medial temporal lobe of the diencephalon. This information is retained by conscious effort and new associations [21]. The acquisition of explicit memory is a three-phase process:Encoding: capturing details about stimuli and the environment for subsequent consolidation (processing the information to store it).Storage: retention of information over time.Retrieval: using the retained information to create a conscious representation of the event or execute a learned motor response.

Explicit memory includes three different categories: immediate memory, short-term memory, and long-term memory [19,20,21] (Figure 1).

## 3. Human Cognitive Assessment

Cognitive assessment is a long and laborious process, whose level of detail depends on the purpose.

A fundamental principle when establishing a person’s cognitive capacity is considering the different cognitive domains [22] (Table 2). These domains correlate with specific brain areas (mainly the associative cortex, subcortical regions, and their networks), establishing a topographical correlation that enables the identification of injured or functionally altered areas in the nervous system [23] (Figure 1 and Table 3). A general way to classify cognitive abilities is to consider the general process involved, such as memory, language, attention, or the brain area involved.

The neurological examination aims to establish the topographic diagnosis in a specific patient. This information, together with the anamnesis, facilitates the preparation of a syndromic diagnosis, a necessary step to reach the etiological diagnosis and, if necessary, to prescribe a treatment.

The approach to cognitive evaluations is complex, requiring the examination of the patient themselves and collaboration with a reliable witness, usually a family member or caregiver, to assess the performance of activities.

### 3.1. Anamnesis

The first step in a cognitive assessment is the interview with the patient, family member, or caregiver. This provides a first impression of the cognitive functions that may be most affected and their impact on functional capacity. This criterion is important because, for the diagnosis of dementia, in addition to verifying the deficit in more than one cognitive function, it is necessary to confirm the change in the performance of activities.

In some cases, with a compatible clinical history, the evaluation may be simple, requiring only a basic cognitive test and some complementary tests to rule out secondary cognitive impairment. Other cases can be more complex, especially in circumstances with psychiatric comorbidities. In each situation, the assessment strategy should be designed to obtain the evidence necessary to determine the patient’s cognitive status. A well-structured anamnesis is the first step to getting an overall idea of the problem and guides the rest of the patient examination. Questions should be specific and explore the patient’s primary functions in daily and instrumental activities.

### 3.2. Physical Examination

Once the anamnesis is completed, the physical examination is performed with a general approach to assess the most affected functions. In many cases, unlike other neurological diseases, the assessment may be normal. However, specific signs suggesting secondary cognitive impairment or a neurodegenerative process should be considered. Some examples of these alterations include the presence of frontal or archaic release reflexes (pacifier reflex, palmomentonian, persistent glabellar), signs of involvement of the extrapyramidal system (pathological eye tracking, tremor, gait disturbance, rigidity, buccolingual dyskinesias, or dystonia), or motor neurons (muscle atrophy, fasciculations, or hyperreflexia).

## 4. Cognitive Function Screening Tools

Three main tools explore the different cognitive functions in routine clinical practice: cognitive tests, functional capacity scales, and advanced neuroimaging studies. Neuroimaging is a tool under development that, thanks to the integration of artificial intelligence, will become an essential method to facilitate the early and objective quantification of deficits [24].

This review will consider cognitive tests, the most commonly used functional capacity scales, and those resulting from previous evidence. On the other hand, it is helpful to differentiate the primary or general tests from the more complex and specific ones (Figure 2).

### 4.1. Basic Cognitive and Functional Tests

Generally, the cognitive examination will begin with a brief screening test, which offers a rapid and global assessment of the patient’s basic cognitive situation [25]. The most common is probably the Mini-Mental Status Exam (MMSE) [26], but others are routinely performed in general neurology consultations and take only a few minutes (Figure 2).

It is essential to consider the patient’s educational level in order to adapt the neuropsychological evaluation in illiterate patients. The most frequently evaluated short tests are the MMSE, the Eurotest, and the Phototest [26] (Figure 2). These basic tests can be supplemented with more targeted tests to consider specific deficits.

There are some shorter scales and tests adapted to the conditions of patients with advanced dementia, such as the Severe Impairment Battery (SIB) and its abbreviated version, the Severe Mini-Mental State Examination (SMMSE) and Severe Cognitive Impairment Profile (SCIP) [25]. The main characteristics of these basic tests are summarized below.

#### 4.1.1. MMSE

The MMSE is the most widely used test. It explores the domains of memory, attention, orientation, language, and praxis, and it has validated versions in many languages [26]. The test takes 7–10 min and has a sensitivity of 88.3% and specificity of 82.6%, with a cutoff point of 23/24 out of 30 total points in the original version. A decrease of three points is considered clinically significant [27].

#### 4.1.2. Phototest

Phototest is a brief cognitive test in which the patient is shown pictures of six objects to be named and recalled. The patient is asked to perform a verbal fluency task between these two tasks. As a result, the test is not significantly influenced by the person performing it. It is not influenced by cultural elements, has good reliability, and is recommended for patients with a low literacy level when the evaluator is not always the same person [28].

#### 4.1.3. Eurotest

This basic screening test should complement other more complete tests. It is based on a calculation in euros and can be adapted to the currencies of many countries. It includes a distraction task naming animals. The maximum score is 35, and the cutoff is 23 [29].

#### 4.1.4. Brief Cognitive Screening Test (Mini-Cog)

Mini-Cog is a psychometric test of memory, executive function, language, and praxis, requiring the patient to draw a clock and recall three words. The drawing of the clock tests the patient’s ability to place all the numbers and hands in the correct position to show the time. The range of scores is from 0 to 5, and it only takes about 3 or 4 min to perform in the office. Sensitivity is 76% to 100%, and specificity is 54% to 85% [27]. This test is suitable for people with a low educational level.

#### 4.1.5. Memory Alteration Test (M@T)

The M@T focuses mainly on memory and is a good screening test for mild cognitive impairment of the hippocampal profile, the most frequent in our environment. Its performance is more complex and explores several types of memory. The assessment takes 5 to 10 min, the maximum score is 50, and cutoff points have been established for Alzheimer’s disease (33 points) [30] and amnesic mild cognitive impairment (36 points) [31].

#### 4.1.6. Montreal Cognitive Assessment (MoCa)

The MoCa is a psychometric test of memory, executive function, language, and praxis. It can detect mild cognitive impairment better than the tests described above. The range is from 0 to 30, it takes about 10 min to perform, sensitivity is 80% to 100%, and the specificity is 50% to 75%, considering 25/26 as the cutoff point [32].

#### 4.1.7. Neuropsychiatric Inventory Questionnaire (NPI-Q)

The NPI-Q is a self-administered test of the caregiver, which takes about 5 min and consists of 12 questions on behavioral symptoms associated with different types of dementia. The score, from 0 to 60, considers the intensity of symptoms and caregiver distress [33].

#### 4.1.8. Functional Activities Questionnaire (FAQ)

This test is completed by the caregiver and elicits information on the patient’s performance of activities of daily living. It ranges from 0 to 30 and takes 5 min. It helps to identify patient risks and areas for improvement in patient care [34].

#### 4.1.9. Lawton and Brody Scale for Instrumental Activities of Daily Living

This scale assesses the patient’s ability to perform eight instrumental activities of daily living (laundry, housekeeping, shopping, meal preparation, use of the telephone, use of means of transportation, responsibility for medication, and management of finances) and measures the patient’s level of independence. It is not specific to patients with cognitive impairment but is useful for functional assessment [35].

#### 4.1.10. Barthel Index

This is the most widely used scale for assessing activities of daily living. It evaluates activities such as the ability to dress, eat, and groom. The result is expressed as a percentage, with 100% indicating complete independence. It is helpful for patient follow-up [36].

It is highly recommended to consider the most suitable test to evaluate the patient and the one in which the evaluator has the experience to guarantee the reliability of the result.

## 5. Advanced Cognitive Evaluation and Specific Tests

For this evaluation, a series of batteries are used, such as the Barcelona Test, Consortium to Establish a Registry for Alzheimer’s Disease (CERAD) [37], Alzheimer’s Disease Assessment Scale-Cognitive (ADAS-cog), Repeatable Battery for the Assessment of Neuropsychological Status (RBANS), and standardized scales that analyze, either generally or specifically, the different areas and neural networks to perform detailed cognitive assessments. These are long and complicated studies that provide much information but may require an effort of attention on the part of the patient that is not always possible to achieve.

In general, test results should be stratified by age and educational level based on standard groups to obtain a cutoff point [25]. An evaluation by a neuropsychological professional is often necessary.

Perhaps the most widely used globally is the ADAS-cog, which is part of the ADAS test that evaluates 11 tasks, including word retrieval, object naming, finger naming, following commands, constructional and ideational praxis, orientation, word recognition, recalling instructions, spoken language, comprehension, and word-finding difficulty. The score range is from 0 to 70, its execution takes about 30 min, sensitivity is 89.1%, and specificity is 88.5%, making it highly reliable.

The exploration of the different cognitive domains is detailed below.

### 5.1. Attention and Executive Function

The attention system is based on the awareness of the cognitive system and the selection or orientation to relevant stimuli (information selection and inhibition mechanisms). The level of alertness involves the ascending reticular formation of the brainstem and the frontal region [38].

The pre-frontal-subcortical network (including the anterior cingulate gyrus) is related to attention and executive function. Selective attention focuses on relevant information and neutralizes irrelevant information. Sustained attention maintains this selective focus while performing a particular task [38].

In the attention test, the level of attention should be explored initially since it interferes with the rest of the cognitive function [25,38]. Assessing factors impairing attention, such as drugs, mood alterations, or fatigue, is essential.

Overall attention can be assessed using orientation tests (temporal, spatial, personal, situational), direct and inverse digits. Direct and inverse digits are a sequence of digits that the patient must repeat in the same or the opposite order as the examiner; a repetition of at least five digits in the same order and four in inverse order is considered normal).

We can also assess specific fields of attention through different tests (Figure 2):Assessment of mental flexibility: a test of execution of alternating sequences (mental flexibility evaluation) such as alternating graphic series, alternating motor series (example, fist-palm-singing sequence), or alternation of two automated series (Trail making test) and controlled verbal fluency (F-A-S).Evaluating the ability to inhibit irrelevant automatisms: go/no-go test, repetition of inverse digits, Stroop test.

The go/no-go test consists of three parts [39]:In the first section, the patient is asked to show one finger when the evaluator shows one finger and two fingers when the evaluator shows two.In the second section, the patient is asked to show two fingers when the evaluator shows one and one finger when the evaluator shows two.In the third section, the patient is asked to show one finger when the evaluator shows two, and none when the evaluator shows one.Assessment of the ability to inhibit impulsivity [39]: impulsivity is expressed as short reaction times and the absence of a work plan. Several tests help us to assess impulsivity, such as the Tower of London test.Assessment of distributed attention through dual-task evaluations [39]. Executive function (abstraction, task planning, organization, sequential execution) can be evaluated by different neuropsychological tests that assess mental flexibility, the ability to inhibit responses, the ability to perform alternating responses, abstraction, and processing speed.Executive function tests: the most widely used are the Luria sequence, the F-A-S, verbal fluency (phonetic fluency is related to prefrontal function [23]), Trail making test part B (sequence letters and numbers), the Stroop test, the Tower of London and Hanoy, the Wisconsin Card Sorting Test (WCST) and frontal assessment batteries, face processing, and body schema assessment (right/left orientation, digital gnosis, etc.).

### 5.2. Perception and Orientation

The perceptual (gnostic) system is in charge of processing sensory information and interpreting what is perceived with the agnosia of each sensory modality [40]. In the evaluation of the visual-perceptual function, we can assess: the processing of sensory information, the integration of the basic characteristics of the stimulus (for example, matching identical figures), object recognition (incomplete object drawing test), and identification (semantic image matching), face processing, and evaluation of the body schema (right/left orientation, digital gnosias, etc.). We can also evaluate the visual-perceptual and visuospatial function (perception of object location, movement, and spatial sphere) using batteries of tests (Figure 2). We can use tests of copies of figures such as the intersection of pentagons, the complex figure of Benson, or the drawing of a clock [23].

### 5.3. Praxias

The primary brain location in charge of praxis is the parietal-temporal region (mainly left). This concept implies processing information to elaborate a motor plan or voluntary, learned, and purposeful movement program. In addition to the praxis system, other systems include the visual-perceptual, language, and symbolic systems. The motor programs of the praxis can be acquired through systematic or incidental training.

The multiple praxes (gestural, constructive) may be evaluated through different tasks (simple repetitive movements, unfamiliar gestures, gestural sequences, familiar gestures not related to the use of objects, use of present or imagined objects, gestures related to specific body parts, construction tasks, drawing tests) or specific batteries/tests [38,39,40].

### 5.4. Memory

Memory can be divided into:Working memory, short-term memory, and long-term memory. The distinguishing feature of the working memory is that it uses executive functions, whereas short-term memory requires neither attention nor active maintenance.Declarative memory or non-declarative memory.

Declarative memory is subdivided into:Episodic (or autobiographical) memory. It depends mainly on the hippocampus, entorhinal cortex, amygdala, and diencephalic structures.Semantic memory. It is associated with the lateral temporal associative cortex.

We can assess verbal and visual memory [25]. The evaluation of declarative memory is carried out with different neuropsychological tests and batteries: word list, free and facilitated recall test, M@T, Rey–Osterrieth complex figure (copy that also serves to evaluate visuospatial domain and subsequent drawing with free recall), memory impairment screen (MIS), Wechsler memory scale, California Verbal Learning Test (CVLT) or Verbal Learning Test Spain—Complutense (TAVEC), and the Face–Name Associative Memory Exam (FNAME) (Figure 1).

The Rey–Osterrieth complex figure consists of copying and delayed reproduction of the figure (complex geometric figure). The strategy used in the copying and the time required for its completion are evaluated. This way, we can assess the visual-perceptual and visual-constructive function, planning, and immediate and delayed recall. Non-declarative or procedural memory comprises skills, cognitive, behavioral, and motor schemes. The networks involved in this memory are diffuse, including orientation and perception networks, parietal and temporolateral cortex, and mainly basal ganglia [41,42].

Access to stored information can be performed implicitly non-consciously (as in procedural memory) or explicitly consciously (as in declarative memory) [43,44].

The assessment of procedural memory is usually evaluated in the context of praxis and problem-solving.

### 5.5. Language

The network involved in the language is in the left hemisphere, distributed in the frontal, parietal, and lateral temporal lobes. It is a complex function that includes understanding and producing language, accessing semantic memory, identifying objects by name, and responding to verbal instructions and specific behaviors. It implies the interaction of several higher functions or cognitive domains [22].

Fluency, naming, comprehension, and repetition should be assessed in language screening (Figure 2). The most used specific language tests are Boston Diagnostic Aphasia Evaluation (BDAE), Boston Naming Test, Western Aphasia Battery (WAB), Psycholinguistic Assessments of Language Processing in Aphasia (PALPA), Mississippi Aphasia Screening, Spanish version (MASTsp), and Token test, among others.

The Boston Naming Test is one of the most widely used tests for assessing aphasia in clinical practice. It consists of 60 sheets with black and white drawings. The test evaluates the patient’s ability to name the drawings. The Boston test is an extensive test for diagnosing aphasia that assesses language in comprehension, repetition, reading, naming, reading comprehension, and writing.

## 6. Cognitive Assessment in Special Populations

It is relevant to consider the patient’s age, cultural background, educational level, and other individual factors since certain diseases may make conventional tests unsuitable for the psychometry of patients. For instance, assessing patients with a concomitant psychiatric disease is complicated.

### 6.1. Cognitive Assessment of the Very Elderly Population

The prevalence of cognitive impairment and dementia in older people is higher since age is the leading risk factor. The evidence for the reliability of mild cognitive impairment and the treatment benefit/risk ratio is reduced. On the other hand, the sensitivity and specificity of cognitive tests are reduced [45], and the side effects of the drugs used to treat these diseases (central acetylcholinesterase inhibitors and NMDA receptor antagonists) are significant. Therefore, special care should be taken using emphasizing scales that measure functionality and evaluate the outcome of interventions.

### 6.2. Cognitive Assessment in Low-Literacy Patients

The usual cognitive assessment tests require a minimum level of literacy skills. Especially in developed countries, several attempts have been made to validate these tests. For some tests, there are studies of acceptable quality. This would be the case with the MMSE, the Cognitive Abilities Screening Instrument (CASI), Eurotest, and Fototest, although none can be recommended as a reliable screening tool [46]. It is more advisable to use, also in that case, tests that assess functional abilities in the performance of activities of daily living or instrumental activities.

### 6.3. Cognitive Assessment of Patients with Advanced Dementia

In patients with advanced stages of dementia, basic and specific tests are not valid because of their low sensitivity to detect changes in low score ranges. There are adapted tests for this type of population, such as Severe Impairment Battery, Severe Cognitive Impairment Battery, and Severe Mini-Mental State Examination [47]. Therefore, it is essential to substitute the usual tests for these adapted ones and select the ones that are the quickest to perform.

### 6.4. Selective Neuropsychological Studies in Psychiatric Populations

Neurocognitive studies can be difficult in the psychiatric population. Patients with schizophrenia present cognitive impairment in attention, verbal and spatial memory, working memory, executive function, and processing speed, as do patients with major depression or bipolar disorder. The latter group may show impaired attention, processing speed, executive function, and episodic memory [48].

Several scales assess the functional level of patients with psychiatric diseases, such as the UPSA scale (University of California San Diego Performance-Based Skills Assessment) and its abbreviated version. However, given the characteristics of this population, it can be challenging to discern what part of the functional impact is secondary to cognitive impairment and what stems from the psychiatric disorder itself [48]. Specific tests have been developed for the cognitive study in this population. At the screening level, one of the scales is the SCIP—Screen for Cognitive Impairment in Psychiatry, which evaluates verbal memory (immediate and delayed, working memory, verbal fluency, and processing speed [49,50] and has been validated in different psychiatric disorders. Among the tests, the BACS—Brief Assessment of Cognition in Schizophrenia evaluates verbal memory, working memory, fluency, attention, processing speed, and executive function [48].

## 7. Experimental Animal Assessment

Learning and memory processes have been studied over the years using animal models, which have provided insight into the brain structures involved in their functioning.

### Types of Animal Models of Cognitive Disorders

Animal models of cognitive disorders have been mainly based on primates, rats, and mice, whose neuronal processes are in several aspects similar to human cognitive functions. Several types of experimental models can be distinguished:Pharmacological models: allowing the evaluation of drugs.Genetic models: based on genetically manipulated animals (transgenic and knockout) [51,52,53,54,55,56].Toxicological models: to determine the toxicity of heavy metals, toxicants, and neurotoxins [57].

Animal models simulate specific syndromes of cognitive deficit in which the triggering factor is aging [58,59,60] or cranioencephalic trauma [61,62]. Recently, other complementary models using non-mammalian animals, such as zebrafish (Danio reiro), vinegar flies (Drosophila melanogaster), and worms (Caenorhabditis elegans), have also been used to screen for potentially toxic or therapeutic compounds and to determine some of the molecular basis of cognitive function.

## 8. Behavioral Tests for Evaluating Cognitive Disorders in Animal Models

The study of cognitive function using animal models includes multiple dimensions, and it is relevant to decide which of them to investigate in order to select the most appropriate specific tests (Table 1).

### 8.1. Assessment of Attention

#### 8.1.1. Pre-Attention: Prepulse Inhibition

This test assesses an individual’s ability to filter the available information by applying a low-intensity stimulus (prepulse) preceding an intense stimulation (pulse). If a lower-intensity stimulus precedes a loud stimulus, the response to the second stimulus is reduced. A neutral (white) sound is constantly maintained to create a stable background noise in this test. The prepulse acoustic stimulus is applied between 4 dB and 18 dB above this background noise. It systematically precedes the larger stimulus (pulse), which varies in magnitude between 100 dB and 120 dB and is applied for 60 ms to 140 ms. The initial auditory prepulse inhibits the magnitude of the subsequent startle response to the pulse, an inhibition that increases as the prepulse intensity increases [62].

#### 8.1.2. Attention

Most theories in cognitive psychology describe at least three distinct types of attention. Sustained attention or vigilance is attending to one stimulus over a significant period. Selective attention focuses on one stimulus. Orientation attention is the directional or spatial orientation, and divided attention is simultaneously attending to two or more different stimuli or performing multiple tasks [63,64].

##### Sustained (Vigilance) and Selective Attention: Five-Choice Serial Reaction Time Task (5-CSRTT)

Trevor Robbins and colleagues developed the 5-CSRTT in 2002 to measure the effects of systemic drug treatments [65]. Its use has been extended to mice and primates, highlighting its translational value. Indeed, it is the preclinical analog of the continuous performance test (CPT). This task is frequently used to evaluate sustained and selective attention [66]. It is also used to assess impulsive behavior or response inhibition. In the 5-CSRTT, mice or rats are trained to respond to a brief stimulus (light) presented unpredictably in one of five locations. Once trained to stable output levels, it assesses the subject’s ability to spatially divide its attention across multiple signal locations to select the correct target stimulus, which produces a food reward. Response speed and choice accuracy are measured and related to attentional performance. This behavioral task measures attention to multiple locations over time [67,68] (Figure 3).

The apparatus is an operant conditioning chamber. The front has five nose poke openings with a light in each one. The aluminum back wall has the food magazine connected to a pellet dispenser. The trainer takes several weeks and begins with intertrial intervals (ITI). At the end of ITI, a light is presented in one of the openings. If the animal goes to the light, a food pellet is delivered to the magazine, and the magazine light is illuminated. If the response is incorrect, omission results in a timeout for 5 s. The accuracy of the answers is calculated, so that the number of correct answers over the total number of responses is a measure of attention [69].

On the other hand, the different parameters of the task can be modified depending on whether we want to enhance sustained or selective attention. For instance, if we increase the temporal unpredictability of the stimulus presentation, we improve sustained attention. In contrast, if we increase the number of times the stimulus location is repeated, we enhance selective attention. [70].

##### Go/No-Go Test: This Test Is Divided into Go and No-Go Tests

The go test presents a stimulus (e.g., light) to the experimental animal, to which the animal must respond and receive a reward. In the case of the no-go test, a specific signal is applied before the go signal (e.g., a tone before the light), and the experimental animal must learn to withhold its response to receive the reinforcement (Figure 4).

The test consists of three phases:Training: the mouse learns to perform a task to obtain reinforcement, and after this, the go cue is applied for five days, in which 40 trials per day are performed.Go test: begins with the presentation of the go signal, and when the mouse responds to the stimulus within a certain period (e.g., 60 s), the reward is delivered. The mouse then has a set time, 20 s, to locate and consume the food. When the mouse completes 85% of the go trials for at least three consecutive days, it enters the go/no go test. Failure on any part of the test causes interruption before the next no-go test.Go/no-go test: this phase lasts approximately 10 days, during which go and no-go tests are randomly interspersed. After the no-go signal, the mouse must learn to hold its response for about 15 s to receive the reward. If the mouse fails to delay its response for a given period (15 s), the test should be repeated with a shorter time interval (2–10 s) to determine whether the deficit is due to an inability to delay its response for a prolonged period or to a failure to learn the no-go task. If the mouse responds prematurely or while the signal is being applied, it is considered a failure, and the reward is not provided.

##### Selective Attention: Latent Inhibition Test

Latent inhibition refers to the delay in conditioning a stimulus when that stimulus has been previously presented without any consequence [59]. This test allows for studying the mouse’s selective attention and specific aspects of learning and memory. The procedure involves two groups of animals; the first group is placed in the apparatus where the test is performed and allowed to explore it for one hour. The second group is placed in the device simultaneously but receives the conditioned stimulus (e.g., a flashing tone and/or lights). The next day, the mice are tested in the same apparatus, and the conditioned stimulus precedes an electric shock to the paws (aversive stimulus). If the mouse leaves the chamber during the 8 s tone, it is considered an avoidance response. If it leaves the chamber after the 8 s tone and during the presentation of the following 8 s tone and electric shock, the response is recorded as an escape. The response is considered a failure if it does not leave the chamber at any of the above times. With this system, learning curves are achieved in which mouse avoidance, escape, and failed responses can be followed in detail. When the two groups are compared, apparent differences between them are observed. Latent inhibition is evident in the mice previously exposed to the stimulus, as they delay acquiring the conditioned response.

##### Orientation Attention: Orientation and Habituation

This technique measures the visual orientation response in rats and mice. It involves placing an object in the animal’s field of view and assessing the ability to track the object. The animal is orienting when it directs its head to the object or tracks it while turning. One way to perform the test is to place the animal in a circular area where it is first allowed to undergo a habituation phase of about two minutes, followed by the introduction of a small object that, for 12 s, moves along the perimeter of the circular area in the mouse’s field of view. Each presentation of the object represents a trial of the test. The orientation response is measured at one- to two-second intervals during each trial.

### 8.2. Evaluation of Learning and Memory

#### 8.2.1. Non-Spatial Memory

##### Object Recognition Test

Since Ennaceur et al. [71] introduced the object recognition test, its use as a research model and powerful experimental tool to assess the effects of drugs on memory and neurobiological mechanisms related to learning and memory has been increasing [38,39,40,41,42,43,44,45]. This test is based on the natural tendency of mice to explore new objects and environments and compare them with familiar ones. By exposing the mouse to a series of objects, the mouse is expected to explore them. The exploration time is when the animal sniffs or touches the object with the front paws at a distance less than or equal to 1 cm. In each part of the test, the objects are changed. In the beginning, the time interval between the initial exposure and the following ones is established, thus assessing short and long-term memory. The test is performed in an open field in an evenly lit room with objects of similar texture, color, and size, but of different shapes (Figure 5).

In the first habituation phase, the animal is placed in the open field without any object. A variable time (10–30 min) is allowed to elapse to become familiar with the environment. After 24 h, training is performed, during which two identical objects (objects A) are placed in different positions (position 1 and position 2), and the animal is left inside the box for 5 or 10 min, during which the exploration time (T) is counted. In the next phase, object A (which we will call a familiar object) is left, and a new object (object B) is placed in the other position. Short-term memory is measured one hour and three hours after training, noting the exploration time of the familiar (TA) and novel (TB) objects. Twenty-four hours after training, long-term memory is measured, for which object B is replaced by a novel object (C) and the same procedure is followed. The expected result is that mice without cognitive impairment explore the novel object more than the familiar one. In contrast, for a mouse with some impairment in cognitive function, the familiar and the new object will seem equally novel, and there will be no difference between exploration times.

##### Social Recognition Test

The social recognition procedure was introduced by Holloway and Thor [72] and subsequently developed by Robert Dantzer et al. [73] to study neuropeptidergic regulation of behavior and memory in rats and mice. The social recognition test focuses on the degree of familiarity between two individuals. As described in Figure 6, it is performed in a cage with a window covered by wire mesh [74]. Next to the cage where the test is performed, there is another circular cage, divided into two chambers with a window covered by wire mesh. The test begins with habituation, placing the experimental mouse in the main cage for 10 min to familiarize it with the environment, with the circular chamber remaining empty. In 5 min sessions, the circular box is rotated to confront the experimental mouse in the main box with each of the two compartments. Subsequently, a mouse is placed in one of the compartments of the circular chamber, which we will call “familiar” since it was in contact with the experimental mouse for five min (a mouse from the same cage can be used). A new mouse, which has never been exposed to the experimental mouse, is placed in the other compartment. The experiment is performed four times. In the first three, the animal is confronted with a familiar mouse, and during these sessions, the mouse becomes accustomed to this exposure and varies its social investigation (habituation). The camera is turned around in the fourth exposure, and the new mouse is shown (dishabituation). In all sessions, the exploration time, the time the animal spends approaching or sniffing the mating window, is measured. Mice with unimpaired cognitive function will spend more time sniffing novel mice than familiar mice; those with a cognitive deficit will not distinguish between familiar and novel mice (their exploration time will be similar).

#### 8.2.2. Spatial Memory

##### Morris Water Maze

The water maze, designed by Richard Morris [75], is one of the most commonly used tests to study spatial memory. The test is in a circular pool divided into four imaginary quadrants (northwest, southwest, northeast, and southeast) with a hidden and partially submerged moving escape platform. The mobile platform is in one of the quadrants and should be 0.5 cm below the water. The escape points will be located at seven different equidistant positions, set randomly throughout the test, and the same ones should be used for all animals. Numerous visual cues are placed around the pool to facilitate the animal’s spatial orientation. Light signals are the most used in this type of maze (Figure 7).

In the first acquisition phase, the animal is introduced with its nose pointing to the pool walls to search for the platform for 60 to 120 s. The animal is placed on the platform for 20 to 30 s if it does not find it. An animal is considered to have found the platform when it remains on it for 5 to 10 s. The animal is removed from the platform and rests for four min before starting the subsequent trial. This procedure is repeated with each animal for up to 12 consecutive trials. The animal’s ability to efficiently locate the platform depends on cues surrounding the pool (at least two cues outside the maze are necessary to identify the invisible target). This acquisition or learning phase may last several days. Two days after the end of the previous phase, a final retention test or research trial is conducted without a platform for 60 to 100 s. If the animal has learned, it will swim longer in the target quadrant, i.e., where the platform was previously located. The most common behavioral measures are the escape latency on acquisition trials (time to reach the platform) and the percentage in the target quadrant during the final test. The acquisition is reflected in the lowest escape latencies across days and the highest time rate in the goal quadrant.

##### Barnes Maze

This test allows the study of spatial learning and memory, working and reference memory, short- and long-term memory, and other more complex tasks in rats and mice. The Barnes maze was designed by C.A. Barnes, Professor of Psychology at the University of Arizona [76] and consists of a dry maze where light and noise are used as aversive stimuli. The animal is forced to escape from these stimuli. The maze consists of an opaque plastic disk raised off the ground by a tripod (Figure 8). Holes with a diameter of 25 cm are distributed along the perimeter of the disk, and an opaque plastic escape box is placed under one of them. Various spatial cues can be placed around the cylinder, which will remain constant throughout the study, and false drawers can also be placed under some holes, in which it seems possible to get inside, but whose available space is so small that it prevents the animal from hiding. A light bulb (about 150 W) and a noise generator (about 80 dB) are the aversive stimuli.

The procedure begins with training, which consists of placing the animal in the escape box and leaving it there for 1 min, to continue 1 min later with the different exploration sessions. All sessions begin by placing the animal in the maze’s center inside a “starting chamber,” consisting of an opaque cylinder about 10 cm high. After 10 s, the starter chamber is removed, the light and buzzer are switched on, and the animal can explore the maze. The stress these stimuli produce will induce the animal to seek an enclosed hidden space. The session ends when the animal finds the escape tunnel or when five minutes have elapsed and the animal is manually introduced into the tunnel. When the animal enters the escape chamber, the aversive stimuli cease, and the animal is left in the dark for 1 min before being returned to its cage. The escape tunnel is always located under the same hole, randomly chosen for each animal. The test is performed over several days until there are fewer than three errors in seven or eight consecutive trials. Spatial learning and memory are assessed by measuring the time the animal spends finding the shelter box and the number of errors before finding it. Each time the animal tries to enter a hole that is not the correct one is considered an error. Successive attempts in the same wrong hole are considered the same error.

##### Object Location Test

The object location test (OLT) is a simple and effective test measuring hippocampus-dependent spatial memory in mice, rats, and zebrafish [71,77,78,79]. This test has been widely used to develop different pharmacological or physiological procedures to evaluate the involvement of specific genes in this type of memory and learning process [80,81,82]. The performance is like the NOR test, with some critical differences. The animal is invited to explore a familiar object located in a new place in the exploration area of an evenly lit room. The experiment is performed in several steps (Figure 9). Firstly, the mouse is exposed to an open field to explore it, habituate it to the environment, and reduce any potential stress during the test. After this habituation phase, two identical objects are introduced, and the mouse is placed in the open field to explore these objects with spatial environmental cues. The exploration time of each object is recorded. The mouse is removed from the arena, and one of the objects is moved and placed in another place in the open field. After a delay period, mice are re-exposed to the arena, allowing free exploration and recording the exploration time of both objects. Usually, mice prefer novelty, and if they remember the previous location of the moved object, they will spend more time exploring this object than the other one. However, if there is any cognitive hippocampus-dependent dysfunction, the time for exploring both objects will be the same.

### 8.3. Working Memory

The concept of “working memory” was formulated based on the experiments of David Olton and Werner Honig in the 1970s. Olton and Samuelson [83] developed a device to assess memory in the rodent: the radial arm maze. They considered working memory the animal’s ability to remember which arms it visited in a session. Dudchenko defined working memory as “short-term recall of an object, stimulus, or location used within a testing session, but not between sessions” [84]. Adaptations of spatial learning tasks, such as the radial arm maze (RAM) or Morris water maze (MWM), are often used to assess working memory in chronically stressed animals [85].

Delayed alternation is the tendency of rodents to choose alternative arms or locations of the maze when re-exposed to a device; animals must remember their initial response to select this alternative response. The delayed spatial alternation task using a T-maze and a Y-maze is the most used method for testing working memory in rodents and provides a measure of short-term spatial working memory.

#### 8.3.1. T-Maze

The rodent is first placed at the base of the T, runs up the stem, and enters one of the arms of the T. Here, the rodent may obtain a reward at the arm’s end. The rodent is picked up by the experimenter and replaced at the base of the T. The animal will run up the stem and enter the arm of the T it had not entered on its first run to receive the reward. Correct arm entry results in a reward, whereas incorrect arm entry results in the absence of a reward. A delay period can be introduced between the first and the second run, during which the animal is prevented from entering either arm. Lengthening or shortening the delay period may modify the task difficulty [84,86].

#### 8.3.2. Y-Maze

This test assesses short-term working memory in mice, measuring spontaneous alternation. An enclosed Y-maze apparatus is used to invite mice to explore all three arms and is driven by the innate curiosity of rodents to explore the previously unvisited arm [87]. Mice are placed in the central area and are left to explore it. The number of arm entries and alternations between arms were recorded to calculate the percentage of the alternation behavior. An alternation is defined as consecutive entries into all three arms. A high alternation rate is defined as a high proportion of entries into consecutive arms, involving a standard working memory. In contrast, a low percentage of alternation, defined as a higher proportion of repeated entries into the same arm, is observed in rodents with impaired working memory [88,89,90] (Figure 10).

#### 8.3.3. Eight-Arm Radial Maze Test

Olton and Samuelson developed this test in 1976. It consists of an octagon-shaped central field with eight radially ejected arms. The maze usually has a dark color. At the tail of each arm, a food container is designed so that food contents would not be visible from the central platform. The animal must return to the central platform after inspecting the food at the end of each arm.

The animal must first get familiar with the environment for three days. The animal is placed gently into the center of the maze and is permitted to search around the maze for 15 min [91]. Training occurs once or twice a day for 5 to 8 consecutive days. At the end of each session, the animal searches all eight arms of the maze for 10 min. The total number of entries are analyzed [91]. If working memory is assessed, all the containers in the arms are full of food, and the animal must search each arm only once. If working memory is impaired, the animal will search an arm more than once. If reference memory is assessed, food is placed in only some of the arms, and the animal should only enter those arms. Entering an arm without food indicates impaired spatial memory [91]. Interestingly, C57BL/6J mice performed the multiple T-maze tasks better than the CD1 strain [92].

### 8.4. Associative Learning

Associative learning is an adaptive process that allows an organism to learn to anticipate events. An animal’s behavior must be quantified using visual or mechanical measures of a particular response. Fear conditioning to a cue or context is a form of associative learning used in many species, including mice, rats, and rabbits [93].

There are different paradigms to keep in mind for the given research needs. The difference between contextual fear conditioning and cued fear conditioning is that the conditioned stimulus is added to the context in the second. Trace fear conditioning is similar, except the cue is presented for a period and terminated. Then, following a short interval (100 mc to 60 s), an aversive stimulus is presented.

#### 8.4.1. Contextual Fear Conditioning

Fenselow first described the contextual fear response as a method to evaluate aversive learning and memory using modified Pavlovian conditioning [94,95]. Contextual conditioning consists of taking an animal and placing it in a new environment, providing it with an aversive stimulus, and removing it. When the animal is returned to the same environment, it will generally show a freezing response if it remembers and associates that environment with the aversive stimulus.

Freezing is a species-specific fear response, defined as “no movement except breathing.” This can last from seconds to minutes, depending on the strength of the aversive stimulus, the number of presentations and the degree of learning the subject achieves [96]. Since this discovery, different modalities of contextual fear conditioning have become some of the most widely used methods to evaluate the effect of different pharmacological, physiological, or environmental manipulations on the recall of a traumatic event, which is the application of electric shock. The one warning is that the mice should be calm and healthy before testing. Rodents are exposed to a cage with an electrified grid. After a delayed period, an electric shock is applied, returning the animal to its housing cage immediately after a period. The intensity of this shock varies between studies and authors, but the foot shock is generally 0.17–0.8 mA for 1–2 s. [96,97]. In this case, the experimenter is only interested in observing the fear response. The tone (CS) is not presented in training.

#### 8.4.2. Contextual and Cues Fear Conditioning Test

In this test, the apparatus is a square chamber with an electrified grid floor, a sound source, and a calibrated shock generator [86]. It is used to measure cue learning and contextual learning. In this case, the aversive stimulus is presented at the end of a cue (CS) (light, tone, odor) and is thus paired with the aversive stimulus (US). In the training session, the animal explores the chamber. A tone (auditory) cue of 70–80 dB is presented for 15–30 s. A mild foot shock is administered during the last 2 s of the tone presentation and co-terminates with the tone. The foot shock is generally between 0.17–0.8 mA for 1–2 s. After the shock presentation, an intertrial interval (60–210 s) precedes a second identical trial. Following the final shock presentation, the house light should remain on for an additional 60 s to enable removal of the mouse 30–60 s after the last trial. The day after the training session, the mouse is placed in the chamber and allowed to habituate for 3 min. The same intensity tone cue used in the conditioning session is then activated for the next 3 min. The mouse freezing behavior can be captured live or recorded for later analysis. The test usually takes place over two days.

#### 8.4.3. Step-Down Inhibitory Avoidance

Different researchers described this test in the early 90s [98,99] with some variations, and it is one of the most widely used methods to evaluate aversive memory retention in rodents. In all these situations, the test assesses short- and long-term memory after applying an electric foot shock of low intensity. The test is performed in several steps. Firstly, rodents are placed on a small platform in a cage connected to an electrified grid in the training session. The shock is applied when the animal places its four paws in the grid (0.3–0.5 mA for 2 s). The time it takes the rodent to step down is recorded. The animal returns to its cage immediately after the shock. After the training session, 1 h, 3 h, or 24 h later, the animal is re-exposed to the same cage and placed on the platform. The step-down latency time, the time the animal takes to step down from the platform, is recorded as a direct indicator or the recall of the electric shock application with a cutoff of 180 s. This is one of the most widely used tests for assessing cognitive status in rodents. Aversive memory consolidation processes are involved due to the negative emotional component of applying electric shock. This latency time will be longer in rodents with cognitive impairment than in those with normal cognitive function.

#### 8.4.4. Passive Avoidance Test

The passive avoidance test is a one-way test for studying acquired learning and memory. The animal is conditioned with an aversive stimulus and is subsequently evaluated if it remembers that experience. A box with two compartments, one light and one dark, separated by a sliding door, is used to carry out this test. There is a powerful light focused directly on the mouse in the light compartment, while in the dark compartment, there is a closed circuit that will produce small electric shocks. The mouse is meant to learn to avoid the dark compartment where the shock is applied [99].

The test starts with training that consists of placing the mouse in the illuminated compartment. After 5 s, the door is opened, and the mouse is allowed to go to the dark compartment (the mouse will spontaneously tend to move to dark spaces, as direct light causes anxiety). As soon as the mouse enters the dark chamber with all four paws, the door is closed, and the mouse receives an electric shock of 0.1–0.4 mA for 2 s. Thirty seconds after applying the shock, the mouse is returned to its cage until the test, which consists of putting the mouse back into the illuminated chamber and lifting the door separating the two chambers. The animal is not shocked during the trial and is observed for 5 min, recording various parameters, including the latency time until it enters the dark chamber (as a memory indicator), the number of times it crosses from one chamber to the other, the total time spent in each chamber, and the frequency with which it peers into the dark chamber until it enters it. The test can be conducted 30 to 60 min after training to assess short-term memory or at 8 to 24 h to evaluate long-term memory. Other experimental variants of passive avoidance tests, such as those used in the step-down inhibitory avoidance task, can be used.

#### 8.4.5. Active Avoidance Testing

The active avoidance test can be classified into one-way and two-way active avoidance. The difference is that in the one-way active avoidance test, the mouse always receives the electric shock in the same chamber/location (unidirectional) [100]. In the two-way active avoidance test, the animal learns, through different cues, to predict the electric shock, regardless of the test chamber (context) where it occurs (it is bidirectional) [101]. Active avoidance tests require several learning sessions, unlike passive avoidance tests, in which the mouse learns in a single session. Still, they allow us to examine complex cognitive functions and assess memory acquisition and consolidation in the same animal during the experiment [102].

The one-way active avoidance test is performed in a box with two compartments connected by a door illuminated with natural light. A light bulb is placed in one compartment, and when it is turned on, a sound is emitted for 10 s, and the door is simultaneously opened so that the mouse can change compartments. At the cessation of the light and sound stimulus, the mouse will receive a 0.1 mA shock for 20 s if it has not changed compartment, until it escapes to the adjacent chamber. This test, repeated up to 20 times a day at 20 s intervals, constitutes the training period. The latency time until the avoidance response (escape to the other chamber) occurs is recorded in each trial. Failure is considered if this response does not appear. This daily training is repeated until the mouse reaches a criterion that reflects learning. Learning curves over time reveal the acquisition and consolidation of avoidance.

Depending on the location, the mouse can receive the electric shock on either side in the two-way active avoidance test. The animal must resolve a conflict since the chamber where it received the discharge in the previous trial becomes the safe chamber during the next trial. Therefore, it has to learn to inhibit its tendency to avoid the compartment where it has just received a shock and use the conditioned stimulus to predict and avoid a new shock [101]. This test is performed in a box with two compartments, each illuminated with a light bulb. The mouse is introduced into one of the two compartments and allowed to explore it for 5 min. The sound and light system that constitutes the conditioned stimulus is switched on. An electric shock is applied if the animal does not cross to the adjacent chamber after 8–10 s. The light, sound, and electric shock cease if the animal crosses to the other chamber or if 10 s elapse from applying the electric shock. Responses are coded as avoidance (the mouse travels to the adjacent chamber after the conditioned stimulus but before the electric shock), escape (the animal crosses to the other chamber during the electric shock), or failure (the mouse does not travel to the adjacent chamber during the next 10 s after the electric shock). The learning period consists of performing this test on the same mouse 70 times for 70 min at 30–90 s intervals. Analysis of the learning curves reveals information about the acquisition or loss of each response (short-term memory). Repetition of the test 24 h after the first trial allows us to assess long-term memory.

### 8.5. Emotional Memory

#### 8.5.1. Aversive Conditioning to Taste

The model of aversive conditioning to taste was created in 1955 by Garcia et al. [103]. To develop aversive taste conditioning, the animal must recognize the conditioned stimulus and get sick with exposure to the unconditioned stimulus. This association between the conditioned and unconditioned stimulus must be developed so that the animal avoids it when the conditioned stimulus alone is presented. The test consists of a first habituation phase. The animal is left for 24 h without food. This restriction ensures the subsequent ingestion of an adequate amount, which, in turn, will favor the association between the conditioned and unconditioned stimulus. After 24 h, each mouse is moved to the test cage, and a small feeding trough is placed in the center. Its base can be flavored with 1% vanilla or almond extract and sweetened with 0.25 M saccharin solution. Mice are randomly assigned one of these two flavors. The food bowl should be weighed before and after the conditioning phase (30–60 min) to ensure the animal has consumed enough. After exposure to the conditioned stimulus, each mouse is injected with 0.15 M lithium chloride or sterile water and returned to its cage. It is kept for an additional 24 h with food restriction until subjected to the retention test. This test consists of placing two food devices, one with vanilla aroma and the other with almond extract, for the animal to freely choose. Mice conditioned to vanilla are expected to prefer almonds, while those conditioned to almonds should prefer vanilla.

#### 8.5.2. Fear-Potentiated Overreaction

Fear-potentiated overreaction is a Pavlovian conditioned fear test. A fear response to an acoustic stimulus can be increased when the stimulus is presented with another aversive stimulus that generates fear responses [104,105]. The trial is usually conducted for four days, starting with 5 min of daily acclimatization in the cage where the experiment is conducted. On the first day, fear responses are measured through a range of intense acoustic stimuli, applied for 40 ms at different intensities (100, 105, and 110 dB). It is crucial to test the mouse a few times to avoid frequent presentations that may cause habituation to these stimuli. On the second day, the same stimuli are applied. Still, in half of the trials, the acoustic over-jump stimulus is presented immediately, followed by a 12 kHz, 70 dB tone for 30 s (conditioned stimulus). In the other half, only the acoustic stimulation of the first day is delivered. Calculating the magnitude of the conditioned stimulus’s response to the acoustic stimulus when presented alone is subtracted from the response to the acoustic stimulus when provided with the conditioned stimulus. The resulting value is divided by the number of responses to the acoustic stimulus presented alone and multiplied by 100.

The tone (conditioned stimulus) is applied on the third day, followed by an electric shock (unconditioned stimulus). Positive values indicate that the tone increases the initial startle response; negative values suggest a reduction in the response when the conditioned stimulus is presented. The association between the two stimuli is repeated 10 times, with 90 to 180 s intertrial intervals. After 24 h, the animals are evaluated for fear-potentiated startle under the same conditions used on the second day, measuring the startle potentiation with the same formula used in the preconditioning. In animals that have acquired fear-potentiated startle, the percentage of potentiation should be higher after conditioning.

## 9. Conclusions

Properly assessing memory and associated cognitive alteration helps to tailor treatment in patients with neurodegenerative diseases. Neurological examination and tests facilitate reaching a particular diagnosis and prescribing appropriate treatment to improve the patient’s quality of life.

This review highlights different diagnostic tests that classify patients according to the severity and type of cognitive impairment and the stage of the disease. However, in patients with advanced dementia, basic and specific tests are not valid because of their low sensitivity to detect changes in the low score range.

Animal models of cognitive disorders play a crucial role in understanding the neurochemical basis of cognitive function/dysfunction. With their use, we increase our understanding of the pathophysiological basis of cognitive impairment caused by the simple fact of aging or associated with different clinical conditions, such as Alzheimer’s disease, Parkinson’s disease, vascular dementia, cerebral amyloid angiopathy, prion diseases, amyotrophic lateral sclerosis, Huntington’s disease, schizophrenia, and other neurodegenerative diseases. The translational approach is essential to gain insight into the brain regions that neurodegenerative diseases may impact. Clinical diagnosis alone is often inconclusive; therefore, postmortem anatomopathological studies are necessary to determine the affected areas. Animal models play a vital role in identifying these areas, as they allow for the collection of data that is difficult to obtain through human research. Furthermore, using appropriate animal models helps us identify potential therapeutic targets and test new drugs’ efficacy.

In recent years, algorithms are being developed by means of artificial intelligence, using psychometric, imaging, neurophysiological, and sociodemographic data along with tools such as “supervised learning,” “unsupervised learning,” “deep learning,” or “natural learning processing.” These can enable early identification of brain structures that decrease their activity and correlate them with patterns of function and behavior [24], as well as checking the extent of hypo functioning or injured areas and their progression. It would even be possible to make predictions based on the data referring to a specific patient’s evolution [24]. To date, no studies directly compare the diagnostic reliability between traditional clinical methods and those of artificial intelligence. However, close collaboration will be necessary, and the cognitive assessment of the future will probably integrate this new methodology to benefit patients.

In summary, the purpose of this article was twofold: first, to outline the battery of tests that professionals use to assess cognitive impairment; and second, to discuss the various animal tests that researchers can use to evaluate cognitive impairment in animal models of psychiatric diseases (Table 1). The ultimate goal is to serve as a guide for professionals and researchers seeking information on the main cognitive tests used in clinical and animal behavior for detecting cognitive impairment.

## 10. Materials and Methods

This literature review consisted of an exhaustive search of scientific information in the MEDLINE (PubMed) database. To identify cognitive impairment, the following keywords and their combinations were used: “human” AND “cognitive impairment” AND dementia” AND “animals’ models” To identify methods to measure cognitive impairment, the following keywords and their combinations were used: “attention”, “learning” and “memory”, “cognitive impairment” AND human OR animal models; AND “screening tools” were combined with terms related to the technical approach using the Boolean operator “AND”; “cognitive impairment” and AND humans OR animals’ models. References of identified publications were included in the additional searches. Articles published in predatory journals were excluded in the screening process.

## Figures and Tables

**Figure 1 ijms-24-07653-f001:**
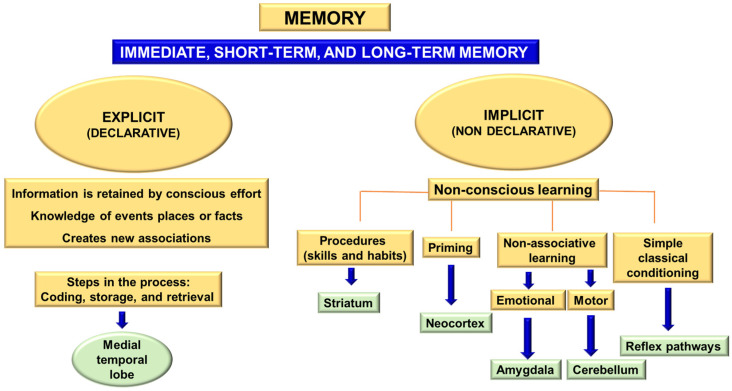
Types of memory and related brain structures. Depending on how the information is received and stored, two main types of memory can be distinguished: implicit memory, which does not require conscious learning, and explicit memory, which is the product of conscious cognitive effort.

**Figure 2 ijms-24-07653-f002:**
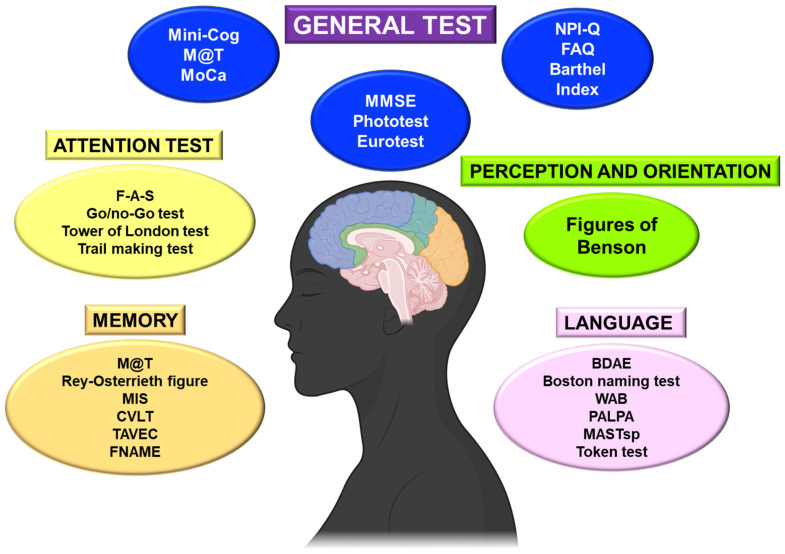
Main tests used to assess cognitive impairment in patients. BDAE: Boston Diagnostic Aphasia Evaluation; CVLT: California Verbal Learning; F-A-S: Controlling verbal fluency; FNAME: Face–Name Memory Exam; MASTsp: Mississippi Aphasia Screening, Spanish version; MIS: Memory Impairment Test; MMSE: Mini-Mental Status Exam; Mini-Cog: Brief Cognitive Screening test; MoCa: Montreal Cognitive Assessment; NPI-Q: Neuropsychiatric Inventory Questionary; PALPA: Language Processing in Aphasia; M@T: Memory Alteration test; TAVEC: Verbal Learning test Spain—Complutense; WAB: Western Aphasia Battery.

**Figure 3 ijms-24-07653-f003:**
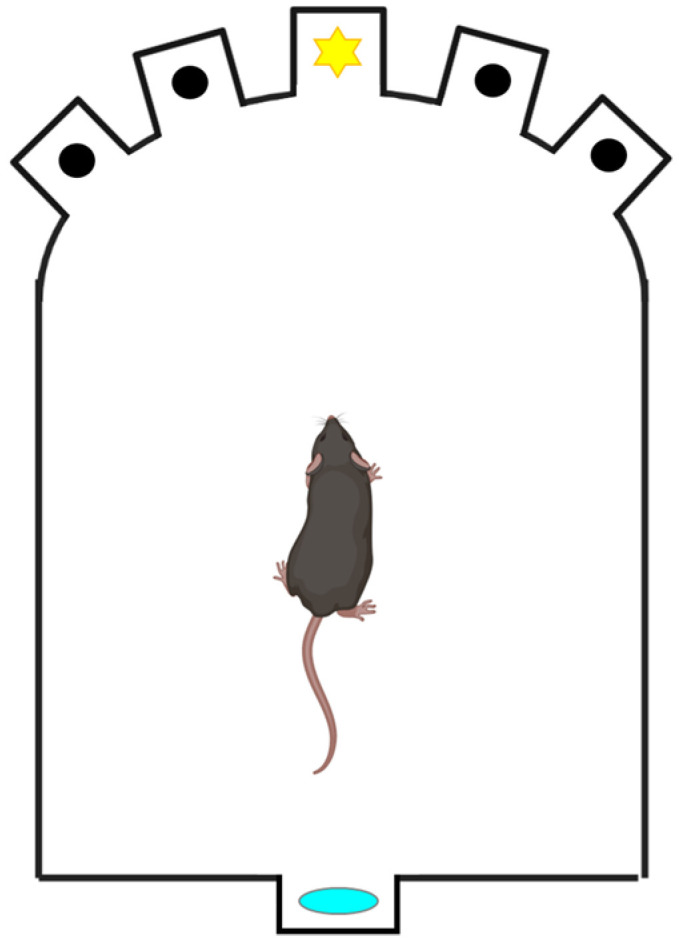
Five-choice serial reaction time task used to assess sustained and selective attention. Rodents are trained to respond to an unpredictable stimulus in one of the five locations. After training, their ability to select the correct target to achieve food reward is evaluated.

**Figure 4 ijms-24-07653-f004:**
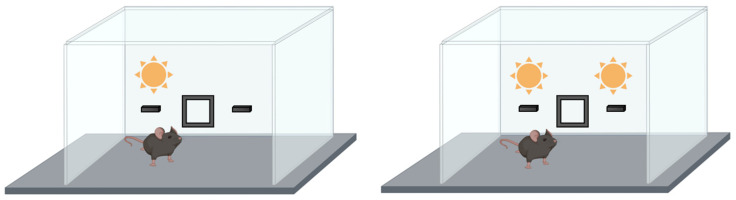
Go/no-go test. This test is performed in operant behavioral chambers where the animal receives go stimuli and must respond by pressing a lever, and will receive a reward, or no-go stimuli, for which it must withhold its response.

**Figure 5 ijms-24-07653-f005:**
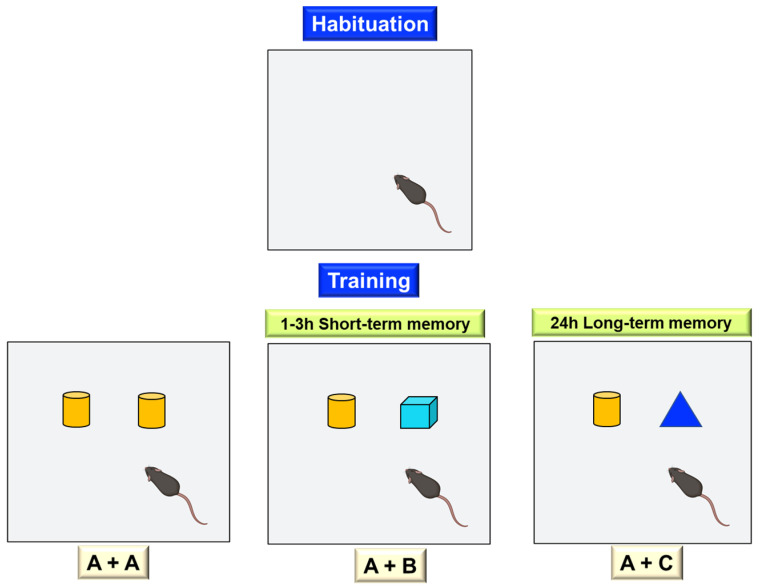
Object recognition test. The experimenter can study short- and long-term memory by modifying the time interval between the first exposure to two identical objects and the following exposures, in which the degree of exploration of the familiar object is compared to a new one.

**Figure 6 ijms-24-07653-f006:**
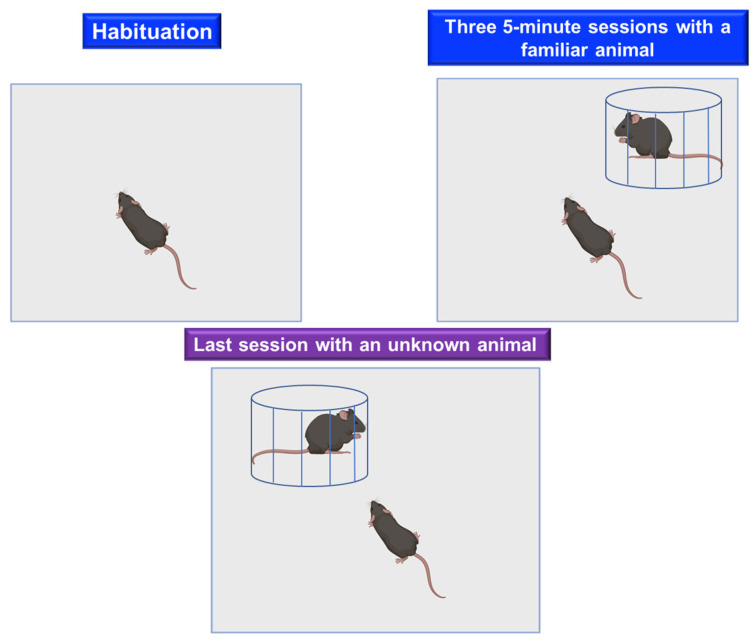
Social recognition test. Repeated exposure of the animal to a familiar individual precedes a final confrontation with an unknown individual, at which time the degree of interaction with the previous one is compared to the degree of interaction with respect to the previous one.

**Figure 7 ijms-24-07653-f007:**
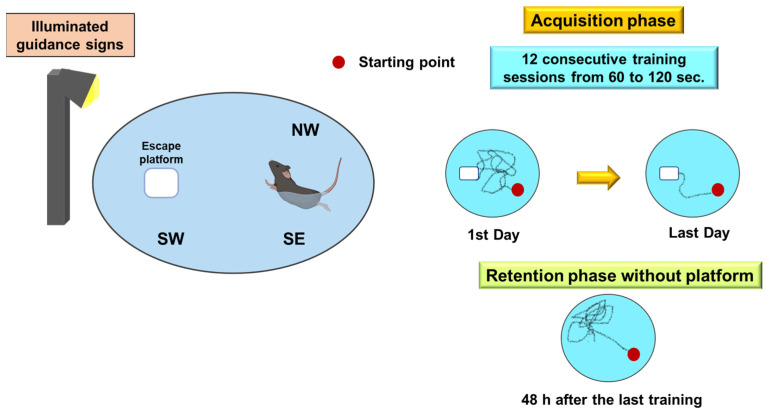
Morris water maze. This test evaluates the latency time that the animal takes to reach the submerged platform, which represents a reinforcing escape stimulus. If the platform remains in the same place, the spatial reference memory is evaluated, and if it varies, the spatial working memory is analyzed.

**Figure 8 ijms-24-07653-f008:**
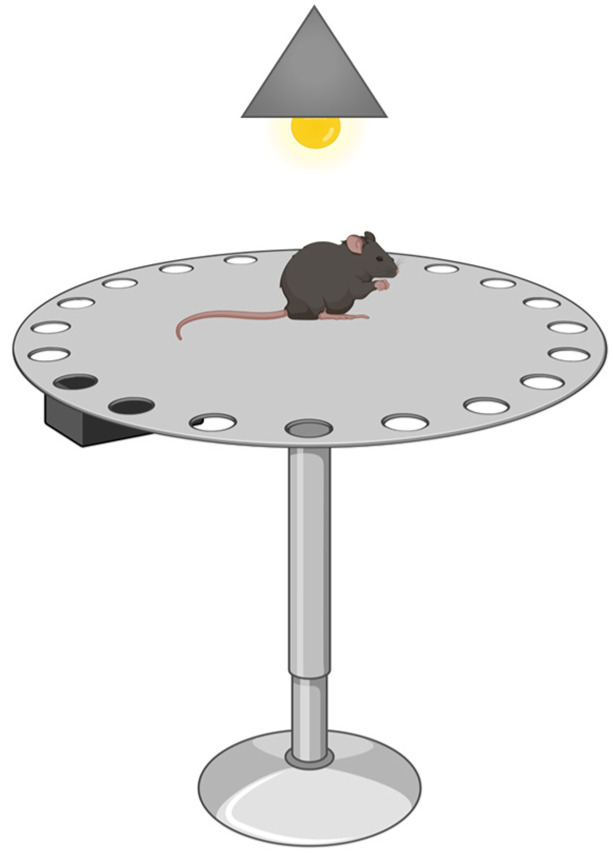
Barnes maze. The animal is first positioned in the center of the apparatus and then allowed to explore the various orifices upon exposure to aversive stimuli, such as lights and sounds. The time it takes to find the exit leading to the refuge box is evaluated.

**Figure 9 ijms-24-07653-f009:**
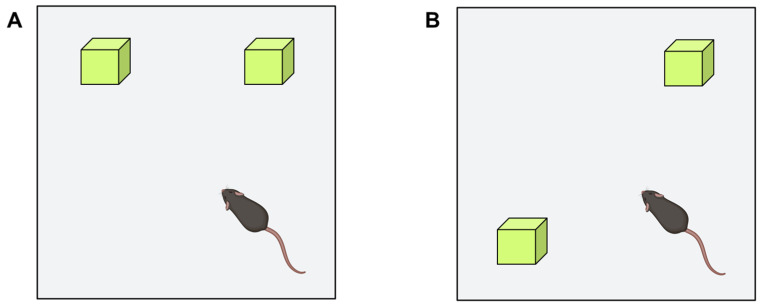
Object location test used to evaluate the spatial memory. During the habituation phase (**A**), mice are invited to explore the two objects located in the open field. After a resting period, mice are re-exposed to the same arena by changing the position of one of the objects (**B**) and measuring the exploration time of each one.

**Figure 10 ijms-24-07653-f010:**
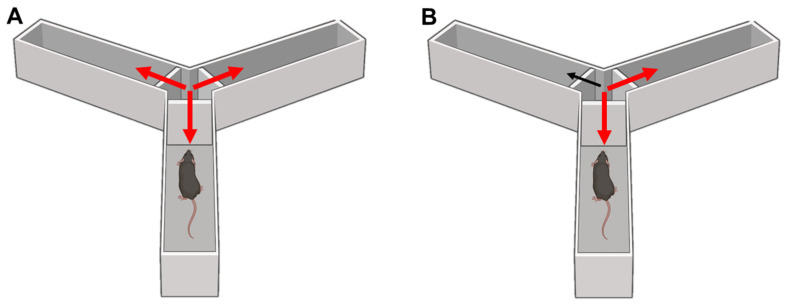
Y-maze test. Working memory is evaluated by measuring spontaneous alternation. In (**A**), red arrows correspond to the normal consecutive exploration of all three arms. In (**B**), the black arrow shows the corresponding arm’s low exploration, indicating spontaneous alternation and working memory problems.

**Table 1 ijms-24-07653-t001:** Summary of the main tests used in human and rodents to evaluate cognitive impairments.

Purpose	Specie	Methods	Specific Field
Evaluation of global assessment of the patient’s basic cognitive situation	Human	MMSE	Attention, orientation, language, and praxis
		Phototest	Verbal fluency
		Eurotest	Complement of other more complete tests
		Mini-Cog	Psychometric test of memory, executive function, language, and praxis
		M@T	Memory
		MoCa	Memory, executive function, language, and praxis
		NPI-Q	Symptoms associated with different types of dementia
		FAQ	Information on the patient’s performance of activities of daily living
		Lawton and Brody scale	Assesses the patient’s ability to perform eight instrumental activities of daily living
		Barthel Index	Assessing activities of daily living
Specific test			
Attention and executive function	Human	Trail making test	Mental flexibility
		F-A-S	
		Go/no go test	Evaluating the ability to inhibit irrelevant automatisms
		Stroop test	
		Tower of London Test	Assessment of the ability to inhibit impulsivity
		Luria sequence	Specific executive function tests
		Trail making test	
		Wisconsin card sorting test	
		Figure of Benson	Perception and orientation
	Rodents	Go/no-go test	Pre-attention
		Pre-pulse inhibition	
		Five-choice serial reaction	Vigilance and selective attention
		Latent inhibition test	Selective attention Orientation attention
		Orientation and habituation	
Memory	Human	M@T	Declarative memory
		Rey Osterrieth complex	Declarative memory and visuospatial domain
		MIS	Verbal and visual memory
		Wechsler memory scale	
		CVLT	
		TAVEC	
		FNAME	
	Rodents	Object recognition test	Non-spatial memory
		Social recognition test	
		Morris water maze	Spatial memory
		Barnes maze	
		Object location test	
		T-maze	Working memory
		Y-maze	
		Eight-arm radial maze test	
		Aversive conditioning to taste	Emotional memory
		Fear potential overreaction	
Learning	Rodents	Contextual fear conditioning	Evaluates aversive learning
		Contextual and cues fear conditioning test	Measures cue and contextual learning
		Step-down inhibitory avoidance	Evaluates aversive memory retention in rodents
		Passive avoidance test	Studies acquired learning and memory
		Active avoidance testing	Associative learning
Language	Humans	BDAE	Fluency, naming, comprehension, and repetition
		Boston naming test	
		WAB	
		PALPA	
		MASTsp	
		Token test	

MMSE: Mini-Mental Status Exam; Mini-Cog: Brief Cognitive Screening Test; M@T: Memory Alteration test; MoCa: Montreal Cognitive Assessment; NPI-Q: Neuropsychiatric Inventory Questionary; FAQ: Functional Activities Questionnaire; F-A-S: Controlling verbal fluency; MIS: Memory Impairment Test; CVLT: California Verbal Learning Test; TAVEC: Verbal Learning Test Spain—Complutense; FNAME: Face–Name Associative Memory Exam; BDAE: Diagnostic Aphasia Evaluation; WAB: Western Aphasia Battery; PALPA: Language Processing in Aphasia; MASTsp: Mississippi Aphasia Screening, Spanish version.

**Table 2 ijms-24-07653-t002:** Cognitive function domain.

Sensations	Sensory Modalities
Perceptions	Object recognition Organizational strategies
Motor and constructive tasks	Copy Drawing Other praxias
Attention and concentration	Selective attention Sustained attention/surveillance
Memory	Working memory: verbal, spatial, object, location Components of memory: central executive, maintenance, manipulation Episodic/declarative memory: Verbal and nonverbal (encoding, storage, free retrieval, cued retrieval, forced recognition by choice) Procedural memory Semantic memory Prospective memory: time-based or event-based
Executive functions	Reasoning Problem-solving Skills management
Processing speed	Semantic fluency Encoding and decoding
Language and verbal skills	Nomination Fluency Reading and comprehension

Adapted from [22].

**Table 3 ijms-24-07653-t003:** Domains to be evaluated and the networks involved.

Cognitive Domains	Neural Networks
Memory	Temporomedial and limbic network: memory and emotional responses
Language	Perisylvian network
Perception	Occipitotemporal network
Praxias	Parieto-frontal network: spatial perception and orientation
Executive-attentional function	Prefrontal-subcortical network: involved in attention and planning

## Data Availability

Data sharing not applicable.

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
