# Peer review of "Methods to Identify Cognitive Alterations from Animals to Humans: A Translational Approach"

_ijms, 2023, doi:10.3390/ijms24087653_

Round 1

Reviewer 1 Report

March 17th, 2023

Dear Dr. Jorge Manzanares, Ph.D.

This manuscript itself is well written, although somewhat descriptive. The authors have conducted a thorough literature review, undertaken a rigorous piece of data collection, and analyzed information accurately. The paper can be accepted as is, with minor grammatical corrections. It is recommended that a native English speaker conduct a minor revision. The authors summarized methods to identify cognition and explained it in detail, especially to confirm the cognitive abilities of humans and animals.

Thank you,

Sincerely yours

Reviewer.

Reviewer 2 Report

This article provided comprehensive reviews on methods to identify cognitive alterations used in humans and behavioral tests in experimental animals to assess a range of cognitive disorders, which summarized the presenting and examining of patterns of cognitive impairment observed in several neurological diseases. The review article is well-organized and provided insights on designing treatment for the patient with a neurodegenerative disease, as well as understanding of the pathophysiological basis of cognitive impairment caused by aging and clinical conditions. It can be accepted for publication as it is.

Reviewer 3 Report

Dear all,

 I found the paper to be overall very well written and I felt confident that you performed careful literature research of the methodologies described here.  I therefore suggest that it may be accepted only after minor revision.

Minor comments:

Line 56-60: the sentence is too long with many "and" that needs to be reworded better.

Line 69: Please write in vitro using italic.

Table 1: please correct the spacing in Memory row

Line 125-126: “Other cases can be complex, especially in cases with comorbidity with psychiatric disease” – to avoid the repetition of the word cases, please replace the second one with especially those.

Line 206-207. Please indicate the cut-off established using TAM for the diseases you exemplify here, as cut-off was indicated above for other methods.

Line 158 you note Figure 1 and at line 273 or 311 is  Fig.1. Please provide constant notations in the text.

Line 281-286: the three section should not have individual bullets. They belong to previous one indicating the citation 32.

For paragraph 4.4 about Memory, I suggest introducing a scheme that exemplifies the text between lines 328-338.

Line 429-448   There isn't much discussion of chapter 7, so I think an imaging figure would help readers who are not exactly in the field to understand the importance of the method. It would be better to place more emphasis on this method's potential.

Line 508: subchapter is 11.1.1 ! The authors should reconsider subchapter numbering in the proper order see line 526 as well.

Line 648: always et al. is italic !

Please introduces spaces before and after all figures!

Line 675. Please indicate the complete name of the scientist; it might be confusing here since there is a Morris syndrome as well discovered by the gynecologist John Morris, who has nothing to do with Moriss who discovered water maze test. You have indicated correctly at Line 709 for Barnes maze, and it is important to keep the content of the information constant, especially since it is a well-documented review.

Even if the length of the article is considerable, I believe that subsection 9 should also brifly include procedures for experimentally inducing a neuropathology in animals, eg. Abeta peptides for Alzheimer or Knockout Mouse Models

Best Regards,

Reviewer 4 Report

The manuscript "methods to identify cognitive alterations from animals to humans: a translational approach" presents a series of behavioural assays for measuring human and animal cognition.

The attempt is not ready for publication but with some restructuring and a table or tables could be publishable.

The abstract is not a summary of the article. The abstract should clearly state what the article is about. Contrary to what is stated in the abstract, the main focus of the article is not about human pathophysiology or about animal models of these disorders. It is a collection of behavioural assays but without the authors indicating the translational aspect. There is no comparison of the tests. It is left to the reader to find out what test to use for an animal model of e.g. Alzheimer disease. 

this can be mitigated by restructuring the article and first presenting the neurological diseases briefly (incl a table) by focusing on the cognitive impairments / cognitive symptoms and then describe the tests used for diagnosing them. As the authors (and reader) will see, there is rarely a specific test for dementia (other cognitive dysfunctions) - which is important to stress. A battery of tests is used - both in humans and in animals. The tests do though differ in how specific they are (as you report for some tests but not all, particularly this information is absent for the animal tests), how easy they are to administer, and importantly for your translational approach - whether they have an animal equivalent. This needs to be made more explicit.

Animals cannot be given the MMSE but there is the SHIRPA protocol https://www.mousephenotype.org/impress/ProcedureInfo?action=list&procID=1376 and https://pubmed.ncbi.nlm.nih.gov/11403965/

see also https://www.ncbi.nlm.nih.gov/pmc/articles/PMC3399545/

The manuscript repeats in section 8 a lot of what is said in section 2

(for inspiration of tables and structure see https://www.ncbi.nlm.nih.gov/pmc/articles/PMC8248664/)

Further major issues
line 280 - I noticed that for a range of tests you cite the same book (ref 32). Please read carefully and find the original descriptions of the tests. It would be helpful for the reader (by e.g. making a table) to compare whether the human test has an animal substitute, i.e. for short term memory or inhibitory control what is used in humans and what in animal models.  

Headings and structure

Heading 5. and 6. should be combined

the heading 7. New tools in cognitive assessment could (if at all) be in the conclusion or outlook section at the very end. It is not part of a translational approach as presented so far

Figure 2 typos. it is explcit and implicit (not explicity and implicity)

the coloring is distracting and not helpful, please remove this

it would be far more useful to compare Fig 1 and 2 in a table to get an overview what kind of tests are used in humans and rodent models for assessing memory / attention / decision-making etc

heading 10. is three lines, then 11. but 10.1.1 - please correct

heading 14 and 15 should be rather one heading and named "fear learning and memory"

The conclusion (like the abstract) is misleading. Here the authors mention the shortcomings of tests for certain populations and then mention brain imaging and animal models. But what is the connection? What can an animal test tell me about a human disease? 

The reader would greatly benefit from learning how animal tests have fed back into human testing, i.e. using non-verbal tests etc. but also that animal models are used for drug testing. 

Materials and methods should come earlier and be more informative. How many articles have been found, how many were in English. Why only ALzheimer and not Lewy bodies as search term? Not also that there are missing " and wrong placed " in this section. When was the search conducted. Since you cite a lot reference 32 - which is a book and or in Spanish (?) - did this also show up in the PubMed search?

Minor issues: 

line 234 Eurotest is described, not fitting to include it in "other basic tests"

line 301 might be a subheading

line 329: STM requires no (very little) attention, WM does. It is the hallmark of WM to recruit executive functions, whereas STM has no active maintenance or attention. It is measured in a similar vein as Tolman identified cognitive maps, i.e. latent learning and surprise recall / recognition tests. 

line 348 Wechsler (not Weschler), and a comma is missing too

line 356 involved in language is in the left hemisphere (no "the" before language) and makes no sense to talk about dominant here

line 401 may result more convenient - that is not English, do you want to say "may yield more convenient results" ? or "may be more convenient" ?

line 450 heading is misleading, The paragraph is not about animal assessment, and should be merged with 2. human cognitive assessment

line 599 which two groups?

line 709 Barnes is a female researcher, she was not designed at the University of Arizona. The task is named after Carol Barnes (whom I met). Please correct the English to avoid misunderstandings 

line 828 including mice, rats, and rabbits (not mouse rats and rabbits)

line 887 - the latency time will be longer, not lower!

there are many English issues (grammar, punctuation) and imprecisions in the description of the tests.

Round 2

Reviewer 4 Report

The article has improved a lot, a few minor issues remain

table 1: Wechsler (you write Weschler)
line 388: "involved in the language is the left hemisphere" should be  "involved in language is in the left hemisphere"

line 431 NeuropsySelective chological study in the psychiatric population should be Selective neuropsychological studies in psychiatric populations 

line 1025ff issues with punctation and "

I find it odd that you only provide figures for the animal test but not the human tests. Since your focus is on helping researchers to test humans, they would benefit from figures of the core tests used in assessing cognitive impairment in e.g. Alzheimer.

Round 3

Reviewer 4 Report

thanks for fixing the minor issues

though it is:

"dummy word", "dummy word" and not as you still have
"dummy word," "dummy word," but the language service at MDPI will fix that for you.